# Genomic Comparison of Two Species of *Samsoniella* with Other Genera in the Family Cordycipitaceae

**DOI:** 10.3390/jof9121146

**Published:** 2023-11-27

**Authors:** Yingling Lu, Zhiqin Wang, Yi Wang, Yue Chen, Dexiang Tang, Hong Yu

**Affiliations:** 1Yunnan Herbal Laboratory, College of Ecology and Environmental Sciences, Yunnan University, Kunming 650504, China; lyinglingua@163.com (Y.L.); w18314560773@163.com (Z.W.); cy106daytoy@163.com (Y.C.); tangdx1516@163.com (D.T.); 2The International Joint Research Center for Sustainable Utilization of Cordyceps Bioresources in China and Southeast Asia, Yunnan University, Kunming 650091, China; 3Laboratory of Forest Plant Cultivation and Utilization, The Key Laboratory of Rare and Endangered Forest Plants of State Forestry Administration, Yunnan Academy of Forestry and Grassland, Kunming 650201, China

**Keywords:** *Samsoniella*, whole-genome sequence, secondary metabolite, biosynthesis gene cluster, gene mining

## Abstract

Whole genomes of *Samsoniella hepiali* ICMM 82-2 and *S. yunnanensis* YFCC 1527 were sequenced and annotated, as well as compared with whole genome sequences of other species in the family Cordycipitaceae. *S. hepiali* ICMM 82-2, *S. hepiali* FENG and *S. yunnanensis* YFCC 1527 had 54, 57 and 58 putative secondary metabolite biosynthetic gene clusters, respectively. *S. hepiali* had one unique domain and *S. yunnanensis* YFCC 1527 six. Both *S. hepiali* and *S. yunnanensis* YFCC 1527 had curvupallide-B, fumosorinone and fujikurin putative biosynthetic gene clusters. *C. javanica* had biosynthetic gene clusters for fumonisin. The 14 genomes had common domains, namely A-P-C-P-C and KS-AT-DH-ER-KR-ACP. The A-P-C-P-C domain may be involved in the biosynthesis of dimethylcoprogen. The maximum likelihood and the Bayesian inference trees of KS-AT-DH-ER-KR-ACP were highly consistent with the multigene phylogenetic tree for the 13 species of Cordycipitaceae. This study facilitates the discovery of novel biologically active SMs from Cordycipitaceae using heterologous expression and gene knockdown methods.

## 1. Introduction

Cordycipitaceae (Ascomycota: Hypocreales) is a family comprising parasitic fungi, including 25 genera such as *Cordyceps*, *Beauveria*, *Akanthomyces*, and *Simplicillium* [1,2]. Among them, *Samsoniella hepiali* (Q.T. Chen and R.Q. Dai ex R.Q. Dai et al.) H. Yu, R.Q. Dai, Y.B. Wang, et al., is an industrial strain for producing Jinshuibao. Its related species is *S. yunnanensis* YFCC 1527 H. Yu, Y.B. Wang, Y. Wang et al. [3]. Based on the current classification position, the genus and species of the related *S. hepiali* are *A. lecanii* (Zimm.) Spatafora, Kepler and B. Shrestha, *C. cicadae* (Miq) Massee, *C. javanica* (Bally) Kepler, B. Shrestha and Spatafora, *C. fumosorosea* (Wize) Kepler, B. Shrestha and Spatafora, *B. bassiana* (Bals.-Criv.) Vuill., *B. pseudobassiana* S.A. Rehner and Humber, *B. brongniartii* (Sacc.) Petch, *Lecanicillium fungicola* (Preuss) Zare and W. Gams and *L. psalliotae* (Treschew) Zare and W. Gams, while the farther genera species apart are *Simplicillium aogashimaense* Nonaka, Kaifuchi and Masuma and *Gamszarea kalimantanensis* (Kurihara and Sukarno) Z.F. Zhang and L. Ca [2]. More than 260 healthcare products had been developed with *S. hepiali* as a raw material, with Jinshuibao capsules having the highest output value [2]. Cordycipitaceae species have produced a variety of metabolites, such as bassianolide [4], fumosorinone [5], ergosterol [6], 8-O-methylindigotide B [7], cordycerebroside B [8], cepharosporolides C [9], beauvericin [10], beauveriolide [11], oosporein [12], bassiatin [13], ophiocordin [14], cordycepin [15], oosporein [16] and cordycepic acid [17]. It can be seen that there is still a lot of room for exploration of the potential of Cordycipitaceae species to produce compounds.

With advances in genome sequencing, many genome sequences have been decoded and made publicly available to mine for the putative secondary metabolites (SMs), and until then, these metabolites have been severely underestimated [18]. The Secondary Metabolites Analysis Shell (antiSMASH) allows for rapid and direct detection of biosynthetic gene clusters (BGCs) as well as the diversity of gene cluster families. These SMs are synthesized by polyketide synthases (PKSs), non-ribosomal peptide synthetases (NRPSs) and terpene synthases (TSs) [19]. Each type I PKS expansion module includes acyltransferase (AT), acyl-carrier-protein (ACP), and setosynthase (KS) domains to achieve a specific cycle of polyketone chain elongation. Some also have domains such as keepeduktase (KR), dehydratase (DH), enoylreduktase (ER), and methyltransferase (MT) to explain the large number of structurally complex and diverse metabolites available through this common pathway [20,21]. The domain of NRPS is usually composed of adenylation (A) and condensation (C) [22,23]. With our current technology, it is very difficult to obtain the TSs.

More SM BGCs can be discovered through gene mining. The amino acid sequence of NpPKS3 in the Lichenized-fungi *Nephrmopsis pallescensi* was 53% consistent with the type III PKS (CYSBs) of *B. bassiana* [24]. Sayari et al. [25] found that all 20 genomes of Ceratocystidaceae showed extremely conserved PKS-III gene clusters containing homologous genes encoding the CHS. Wang et al. [11] discovered beauveriolide BGC in the *C. militaris* (L.) Fr. genome using bioinformatics analysis and then produced the compound via heterologous expression. Fumosorinone was a new 2-pyridone alkaloid isolated from *C. fumosorosea*. The BGC of fumosorinone consisted of a hybrid PKS-NRPS, two cytochrome P450, a trans-enoyl reductase gene, and two other transcription regulatory genes [26]. Xu et al. [27] cloned and catalyzed the synthesis of the bassianolide-encoding gene NRPS (bbBsls) in *B. bassiana* and found that targeted inactivation of bbBsls eliminated the production of bassianolide. The biosynthesis of beauvericin, another cyclic peptide produced by the strain, was not affected. Previously, we conducted comparative genomic studies on the SM BGCs of *Cordyceps* and found that even related species of the same genus had different genome sizes, GC contents and assumed quantities of PKS, NRPS, hybrid PKS-NRPS, hybrid PKS-Other and terpene. Seven compounds, namely dimethylcoprogen, epichloenin A, wortmanamide A/B, phomasetin, fumosorinone, beauvericin, and viriditoxin, have been discovered as putative BGCs [28]. In this study, whole genomes of *S. hepiali* and *S. yunnanensis* YFCC 1527 were sequenced and annotated, and the basic characteristics and gene mining were compared with 14 genomes of 13 species belonging to 7 genera in the family of Cordycipitaceae (i.e., *Akanthomyces*, *Cordyceps*, *Beauveria*, *Lecanicillium*, *Gamszarea*, *Samsoniella*, and *Simplicillium*). The potential of Cordycipitaceae fungi to produce the SMs was further analyzed.

## 2. Materials and Methods

### 2.1. Test Materials

For this study, the strain ICMM 82-2 of *S. hepiali* and YFCC 1527 of *S. yunnanensis* were collected from Yunnan Province in China. The voucher specimens were stored in the Yunnan Herbal Herbarium (YHH) of Yunnan University, and the isolated strains were stored in the Yunnan Fungal Culture Collection (YFCC) of Yunnan University. The genome-wide data of *S. hepiali* FENG, *A. lecanii*, *C. cicadae*, *C. javanica*, *C. fumosorosea*, *B. bassiana*, *B. pseudobassiana*, *B. brongniartii*, *L. fungicola*, *L. psalliotae, Simplicillium aogashimaense* and *G. kalimantanensis* were obtained from the NCBI (https://www.ncbi.nlm.nih.gov/, accessed on 8 October 2022) (Appendix A).

### 2.2. Strain Culture

*S. hepiali* ICMM 82-2 and *S. yunnanensis* YFCC 1527 were cultured on PDA solid medium (20 g/L potato powder, 20 g/L glucose, 18 g/L agar powder, 1 L H_2_O (All chemicals & reagents were from Yunnan, China)) at 28 °C for 15 days.

### 2.3. Genome Sequencing and Assembly

Appropriate amounts of *S. hepiali* ICMM 82-2 and *S. yunnanensis* YFCC 1527 mycelium were scraped, respectively, and the total genomic DNA was extracted using a plant DNA isolation kit (100, Chengdu, Yunnan). Then, it was sequenced on an automatic sequence analyzer (BGI Co., Ltd., Wuhan, China) using the same primers as those used in the amplification. The Illumina NovaSeq 2000 high-throughput sequencing platform was used to construct the *S. hepiali* ICMM 82-2 gene library and *S. yunnanensis* YFCC 1527 gene library with 400 bp insertion fragments (Personalbio, Shanghai, China). FastQC was used for the quality control of the data, and AdapterRemoval (version 2) and SOAPec (v2.0) software were used to filter the raw data to obtain high-quality collocation-free genomic assembly readings. Then, A5-MiSeq and SPAdes were used to construct the contig and scaffold, and the pilon v1.18 was used to evaluate the assembly effects of the contig and scaffold. Finally, BUSCO (Benchmarking Universal Single-Copy Orthologs, http://busco.ezlab.org (accessed on 12 November 2022), v3.0.2) was used to evaluate the integrity of the genome assembly.

### 2.4. Gene Prediction and Annotation

#### 2.4.1. Repeat Sequence Analysis

Homologous annotation and de novo annotation were used to identify the repeat sequences. Homologous annotation was performed using RepeatMasker software (version 4.0.5) [29], with the -species set as fungi and the search engine set as rmblastn2.2.27+. The version of the Repbase database was 20150807 [30]. Comments from scratch were made using RepeatModler (version 1.0.4, http://repeatmasker.org/RepeatModeler.html (accessed on 12 November 2022)) software. 

#### 2.4.2. Prediction of Non-Coding RNA

The tRNAscan-SE (version 1.3.1) was used to predict the presence of tRNA in the whole genome [31]. The RNAmmer (version 1.2) was used to predict the rRNA genes [32]. Predictions of the remaining non-coding RNA were mainly obtained via comparison with Rfam [33]. 

#### 2.4.3. Prediction of Protein-Coding Genes

The Augustus (version 3.03), glimmerHMM (version 3.0.1) and GeneMark-ES (version 4.35) were respectively used to predict the genetic model of this genome from scratch [34,35,36] and to obtain the corresponding gene prediction results. Exonerate software (version 2.2.0, http://www.ebi.ac.uk/about/vertebrate-genomics/software/ (accessed on 12 November 2022)) using the protein sequences of 5 species was used to obtain the corresponding gene prediction results. The EVidenceModeler (version r2012–06-25) was used to integrate the de novo and homology predictions of related species [37]. 

#### 2.4.4. Carbohydrate Active Enzyme (CAZy) Analysis

The hmmscan (3.1 b2, February 2015) software was used to predict the presence of CAZy enzyme genes in the genome sequence. 

#### 2.4.5. Evolutionary Genealogy of Genes: Non-Supervised Orthologous Groups Database (eggNOG) Analysis

The eggNOG -mapper software (version 4.5) was used to complete the work. The database used by diamond was eggNOG (version 4.5, 28 November 2017), and the critical value was selected as 1e-6. The function verification rules of eggNOG were as follows: E-value < 1e-6, select a best hits eggNOG number and assign it to the corresponding protein-coding gene. 

#### 2.4.6. Kyoto Encyclopedia of Genes and Genomes (KEGG) Analysis

The KO and pathway annotations of protein-coding genes were mainly completed using KEGG’s KAAS (version 2.1) automated annotation system [38], where the gene set “For Eukaryotes” was selected. The discriminant rule of the gene KO was the bi-directional best hit (BBH). 

#### 2.4.7. Swiss-Prot Annotation of Protein-Coding Genes

The BLAST software (version 2.2.1) was used to complete the process, and the critical value of the sequence alignment was selected as 1e-6. The functional discrimination rule of the sequence was: E-value < 1e-6, select a best hits Swiss-Prot name and assign it to the corresponding protein-coding gene. 

#### 2.4.8. Gene Ontology (GO) Analysis

The InterPro (version 66.0, release 23 November 2017) software was used to complete the GO annotation of protein-coding genes. 

#### 2.4.9. Cytochrome P450 Analysis

The BLASTP software (version 2.5.0+) was used to predict the presence of P450 genes in the genome sequences. The database used for the sequence comparison was the P450 protein sequence database. The critical value of the sequence alignment was selected as 1e-5, and an optimal number of hits was selected for the functional identification.

### 2.5. Analysis of Secondary Metabolite Biosynthesis Gene Cluster

Gene cluster prediction analysis was performed at the level of 14 genomic scaffolds of Cordycipitaceae species using the online program antiSMASH (https://antismash.secondarymetabolites.org/, accessed on 12 February 2023). Gene structure prediction analysis was performed on the scaffold prediction of the gene clusters via antiSMASH analysis using the online program FGENESH (www.softberry.com/, accessed on 5 March 2023) and using *C. militaris* as a parameter. The contigs located in the NRPS or PKS genes were subjected to gene cluster prediction analysis using the online program PKS/NRPS analysis (nrps.igs.umaryland.edu/, accessed on 18 March 2023) with a view to obtaining the structural domains. The online program Protein BLAST (https://blast.ncbi.nlm.nih.gov/, accessed on 3 April 2023) was also used to perform protein comparative analysis of the contigs in which the NRPS/PKS genes were located.

### 2.6. Synteny Analysis

Scaffolds containing SM BGC from 14 genomes of Cordycipitaceae species were combined in MAUVE v2.4.0 for collinearity analysis in the assembly order.

### 2.7. Cluster Analysis

After comparing the protein or nucleic acid sequences in this study with the known NRPS, PKS, and hybrid PKS-NRPS protein sequences and polygene nucleic acid sequences downloaded from the NCBI, the Clustal W program in the MEGA5.0 software was used for the multi-sequence comparison. Using the maximum likelihood (ML) from the IQ-TREE web server (http://iqtree.cibiv.univie.ac.at/, accessed on 7 May 2023), the quick and accurate phylogenetic trees were constructed via repeating 1000 times. Further the ML clustering trees were constructed using the default parameters, while the multi-gene (SSU, LSU, *TEF*, *RPB1*, *RPB2*) phylogenetic tree and the Bayesian inference (BI) clustering trees containing homologous regions of the NRPS, PKS and hybrid PKS-NRPS were constructed using the software MrBayes-3.2.7.

## 3. Results

### 3.1. Basic Genomic Characteristics of S. hepiali ICMM 82-2 and S. yunnanensis YFCC 1527

#### 3.1.1. Genome Sequencing and Assembly

A total of 20,811,730 raw reads and 20,441,822 high-quality reads of *S. hepiali* ICMM 82-2 were obtained using the Illumina sequencing (Appendix A). *S. hepiali* ICMM 82-2 had a total genome size of 35.92 Mb, containing 645 contigs, 493 scaffolds, and a GC content of 52.85%. A total of 9474 protein-coding genes with a total gene length of 15.06 Mb were predicted for *S. hepiali* ICMM 82-2, along with 111 tRNA secondary structures, 32 snRNA, and 26 rRNA. Its related species, *S. yunnanensis* YFCC 1527, had a high genome assembly quality, with a total genome size of 34.17 Mb, consisting of 275 contigs, 216 scaffolds, and 53.13% GC content (Appendix A). *S. yunnanensis* YFCC 1527 was predicted to have a total of 10,600 protein-coding genes with a total length of 16.94 Mb and a predicted secondary structure of 120, 26, and 32 for the tRNA, rRNA and snRNA, respectively. Meanwhile, the genome assembly integrity of *S. hepiali* ICMM 82-2 and *S. yunnanensis* YFCC 1527 was better, and the complete BUSCOs were 99.6% and 99.7%, respectively (Appendix A).

#### 3.1.2. Genome Annotation

The eggNOG database showed that most of the genes were predicted to have the general function of *S. hepiali* ICMM 82-2 (Figure 1a) and *S. yunnanensis* YFCC 1527 (Appendix A), and the rest were in the “Function unknown” category. Secondly, the “Carbohydrate transport and metabolism, posttranslational modification, protein turnover, chaperones” were the most abundant gene class maps in the eggNOG group. It was indicated that *S. hepiali* ICMM 82-2 and *S. yunnanensis* YFCC 1527 had abundant and diverse carbohydrate metabolic functions and posttranslational events. This may help to improve energy conversion efficiency and regulate protein bioactivity. The results of the KEGG functional classification showed that *S. hepiali* ICMM 82-2 (Appendix A) and *S. yunnanensis* YFCC 1527 (Appendix A) had high activity of protein families, indicating that the two strains had a variety of genetic information, signals and cell proteins, and they might have higher information exchange efficiency and secondary metabolic efficiency. According to the GO annotation results (*S. hepiali* ICMM 82-2: Appendix A; *S. yunnanensis* YFCC 1527: Appendix A), from the cell component category, intracellular and cellular component, from biological processes, cellular nitrogen compound metabolic process and the biosynthetic process, and molecular function from molecular function, ion binding and oxidoreductase activity. It was further indicated that *S. hepiali* ICMM 82-2 and *S. yunnanensis* YFCC 1527, as wild strains, might be related to signal transduction in many metabolic genes.

#### 3.1.3. Additional Annotation

##### Pathogen Host Interactions (PHIs)

Based on the PHIs, the pathogens were mainly fungi, oomycetes and bacteria, and the hosts of infection were animals, plants, fungi and insects [39]. The result of the annotation of the PHIs showed that the major annotated genes of *S. hepiali* ICMM 82-2 and *S. yunnanensis* YFCC 1527 PHI base had decreased virulence and did not affect pathogenicity, indicating that *S. hepiali* ICMM 82-2 (Appendix A) and *S. yunnanensis* YFCC 1527 (Appendix A) were not highly pathogenic strains. The safety of *S. hepiali* ICMM 82-2 as a Jinshuibao capsule was further explained.

##### Carbohydrate Genes

The CAZy were enzymes that played an important role in the carbohydrate modification, biosynthesis and degradation of fungi [40]. It was also a database of carbohydrate active enzymes and a specialized database of carbohydrate enzymes [41]. The results showed that *S. hepiali* ICMM 82-2 (Figure 1b) and *S. yunnanensis* YFCC 1527 (Appendix A) had a lot of glycoside hydrolases (GHs), glycosyl transferases (GTs), and auxiliary activities (AAs), hypothesizing that *S. hepiali* ICMM 82-2 and *S. yunnanensis* YFCC 1527 might have the ability to capture more energy and decompose complex carbohydrates.

### 3.2. Basic Characteristics of 14 Genomes of Cordycipitaceae Species

We found that the assembly quality of the *S. hepiali* FENG genome sequence was better than that of *S. hepiali* ICMM 82-2 (Appendix A). Meanwhile, in a comparison of the genome sequences of 13 Cordycipitaceae species, the highest content of *S. hepiali* FENG GC was 53.90%, indicating that *S. hepiali* ICMM 82-2 and *S. hepiali* FENG belong to different strains of the same species, and their genome sequences were also significantly different. This was primarily reflected in the genome size and GC content. The *L. fungicola* genome had the largest genome size (44.57 Mb), while *Simplicillium aogashimaense* had the smallest genome size and GC content, with 29.25 Mb and 49%, respectively.

### 3.3. Analysis of Secondary Metabolite Biosynthesis Gene Clusters in 13 Cordycipitaceae Species

#### 3.3.1. Overview of 14 Genomic BGCs of Cordycipitaceae Species

AntiSMASH and local BLAST analyses showed significant differences in the number and type of putative SM BGCs in the 14 genomes of Cordycipitaceae (Appendix A). There were 54 and 57 putative SM BGCs in *S. hepiali* ICMM 82-2 and *S. hepiali* FENG, respectively. *S. yunnanensis* YFCC 1527, as a related species of *S. hepiali*, contained 58 putative SM BGCs, including different NRPS, PKS, terpene, hybrid PKS-NRPS, and an indole gene. *S. hepiali* ICMM 82-2, *S. hepiali* FENG and *S. yunnanensis* YFCC 1527 belonged to the genus *Samsoniella*. A homologous gene like chalcone and stilbene synthases (CHS-like) has been found in all three species and *B. bassiana*. *A. lecanii* was a species adjacent to *Samsoniella*. Compared with the genome analysis of the three genomes of *Samsoniella*, there were great differences in the number and type of SM BGCs assumed by *A. lecanii*. *A. lecanii* had 46 SM BGCs and no other gene.

We found that *Cordyceps*, *Beauveria*, and *Lecanicillium*, closely related genera to *Samsoniella*, also differed significantly in the number and type of the putative SM BGCs. The greatest difference was found in the two species of *Lecanicillium*, i.e., *L. fungicola* had the most putative SM BGCs (76) and PKS (31). *L. psalliotae* had 44 putative SM BGCs, including 17 NRPS, 14 PKS (6 HR-PKS, 5 NR-PKS, and 3 PR-PKS), 4 terpenes, 3 hybrid PKS-NRPS, 3 other genes, 1 hybrid NRPS-other, and 1 hybrid PKS-other. There was a small difference between *Cordyceps* and *Beauveria*, but a large difference in the number of BGCs with a putative SM between the three species of *Cordyceps*. *C. cicadae*, *C. javanica* and *C. fumosorosea* had 31, 51 and 54 putative SM BGCs, respectively, including the number of NRPS, PK, hybrid PKS-other and other genes being very different. The three species of *Beauveria* differed greatly only in the amount containing hybrid PKS-other. *Gamszarea* and *Simplicillium*, which were relatively distant relatives of *Samsoniella*, also showed differences between the results of the genomic analyses of these two species and those of the three species of *Samsoniella*. It could be seen that the number and type of putative SM BGC varied considerably between species of different genera and between species of the same genus.

In the analysis of 14 genomes of 13 species in Cordycipitaceae, *C. cicadae*, *C. fumosorosea*, *B. bassiana*, and *B. brongniartii* had no homologous region highly like the hybrid PKS-other. At the same time, similar genetic regions of hybrid PKS-NRPS-other were found in *C. javanica* and *G. kalimantanensis*, and no hybrid PKS-NRPS-other structures were found in the other 12 genomes. These findings further indicate differences between genera, between different species within the same genus, and between different strains within the same species, mainly in the number and type of BGCs with putative SMs.

#### 3.3.2. Difference Analysis of 14 Genomic Domains in Cordycipitaceae Species

The analysis results of the 14 genomes of Cordycipitaceae (Appendix A) showed that there were also significant differences in the structural domains of PKS, NRPS and hybrid PKS-NRPS. The domains A-P-C-P-C and KS-AT-DH-ER-KR-ACP were shared by all 14 genomes, and the A-P-T domain was shared by all 13 genomes except *Simplicillium aogashimaense*. Only *C. javanica* had hybrid PKS-NRPS-other of the KS-AT-MT-P-P-C-A-P-Te-PX-Snx8 domain. Except for *S. hepiali* ICMM 82-2 and *C. fumosorosea*, all the other genomes had the A-C-P-C-P-C domain. Only *A. lecanii*, *C. fumosorosea*, and *L. psalliotae* had not the KS-AT-DH-MT-ER-KR-ACP domain. The KS-AT-DH-ER-KR-P-C-A-P-Te domain existed only in the genomes of *S. hepiali* ICMM 82-2, *S. yunnanensis* YFCC 1527, *A. lecanii*, *C. fumosorosea*, *L. psalliotae*, and *G. kalimantanensis*. 

Each of the 14 genomes had a unique domain. For example, the unique structural domains of *S. hepiali* ICMM 82-2 were the A-C-P-C-P-C, AT-DH-MT-KR, KS-AT-DH and KR-ACP-C-A-Te. The A-P-C-P-P-C, A-P-A-C-A-P, KS-AT-DH-ER-KR-ACP-ACP, KS-AT-DH-MT-ER-ACP-ACP, KS-AT-DH-MT-KR-ACP-C-A-Te, KS-AT-DH-ER-KR-P-C-A-Te, A-P-KS-AT-KR-ACP, KR-P-C-A-Te and A-P-Te-KR were the domains specific to *S. hepiali* FENG, while the domain specific to *S. hepiali* was the SAT-KS-AT-PT-ACP-ACP. *S. yunnanensis* YFCC 1527 was characterized by the C-A-A-A-P-C-A-P-C, C-A-A-P-C-A-Te, KS-AT-DH-MT-KR-ACP, AT-ACP-Aes, SAT-KS-PT-ACP-ACP, and A-P-KS-AT domains.

#### 3.3.3. Analysis of SM BGCs of 13 Species of Cordycipitaceae

SM BGCs catalyzed via NRPS had been predicted in 4 out of 13 Cordycipitaceae species. The key enzymes NRPS (A-P-C-P-C), PTZ00265 super family, Acetyltransferase, PRK08315 super family and MFS super family were known to be required for the synthesis of dimethylcoprogen family working together. By comparison, *S. hepiali* ICMM 82-2 region 25.1, *S. hepiali* FENG region 6.2, *S. yunnanensis* YFCC 1527 region 11.2, *B. bassiana* region 8.4, *B. pseudobassiana* region 8.1, *L. fungicola* region 126.1, *L. psalliotae* region 95.2, *G. kalimantanensis* region 77.1, *C. javanica* region 1.1, *B. brongniartii* region 6.4, *C. fumosorosea* region 75.1, *C. cicadae* region 56.2 and *A. lecanii* region 2.4 all had homologous regions, being highly similar to the synthetic dimethylcoprogen gene cluster (Figure 2a). The slight difference was that there was only the NRPS in *C. javanica* region 1.1 and not the four modified genes mentioned above. *G. kalimantanensis* region 77.1, *B. bassiana* region 8.4, *B. brongniartii* region 6.4 and *A. lecanii* region 2.4 did not contain the modified gene PRK08315 super family. *S. hepiali* ICMM 82-2 region 25.1 had not the MFS super family, and *C. cicadae* region 56.2 and *A. lecanii* region 2.4 did not have the homologous gene of PTZ00265 super family. The gene of FJ439897.1 was a BGC for catalytic synthesis of bassianolide. The results showed that *B. bassiana* region 3.4, *B. brongniartii* region 15.1 and *Simplicillium aogashimaense* region 11.3 might be the BGC of catalytic synthesis of bassianolide (Figure 2b). However, the HET gene was lost in *B. bassiana* region 3.4. *B. brongniartii* region 15.1 only contained C-A-P-C-MT-A-P-C, Abhydrolase super family, CYP_FUM15-like and NAT_SF super family genes. Only the Rab7 and ACAD super family genes were highly similar in *Simplicillium aogashimaense* region 11.3. *L. fungicola* region 241.1 and *A. lecanii* region 4.5 were responsible for the synthesis of acetylaranotin (Figure 2c). *A. lecanii* region 3.1 and *C. javanica* region 18.2 had the potential to be responsible for the synthesis of epichloenin A (Figure 2d). The known gene cluster for catalytic synthesis of acetylaranotin was CH476597.1, whose domain was the P-C-A-P-C, and the key enzyme NRPS in *L. fungicola* region 241.1 and *A. lecanii* region 4.5 was the C-A-P-C. The modified genes GstA, PLN02607 super family, AdoMet_MTases super family and dimerization2 super family involved in the synthesis of acetylaranotin shared a common segment of gene. CYP_GliC-like and TrxB super family shared a gene segment, while these modified genes in *L. fungicola* region 241.1 and *A. lecanii* region 4.5 acted as single genes. It could be seen that the BGC predicting the synthesis of a certain SM had a certain evolutionary relationship among different species of Cordycipitaceae, which was mainly manifested in some structural domains of NRPS, deletion or addition of modified genes, and evolution of modified genes from one gene expressing multiple gene functions to a single gene expressing a single functional gene.

AntiSMASH and local BLAST analyses revealed the existence of eight BGCs catalyzed by PKS, including three HR-PKS, three NR-PKS, one PR-PKS and one HR-PKS adjacent to NR-PKS. Three compounds catalyzed by NR-PKS were citrinin, viriditoxin and bikaverin. *A. lecanii* region 5.5 might be the BGC catalyzed by citrinin synthesis (Figure 2e). *A. lecanii* region 5.5 had more ACP and Aes and lacked some modified genes. *C. cicadae* region 117.1, *Simplicillium aogashimaense* region 14.4 and *C. fumosorosea* region 58.1 might be the gene cluster that catalyzed the synthesis of viriditoxin (Figure 2f). *C. cicadae* region 117.1 had not multi-copper oxidase/laccase gene, and *Simplicillium aogashimaense* region 14.4 had not the ACP structural domain, SDR_c and Abhydrolase super family modified genes. HF679027.1 was a BGC for catalytic synthesis of bikaverin (Figure 2g). Through comparison, it was found that *B. brongniartii* region 2.1 was highly similar to known gene clusters of bikaverin. However, there were not Abhydrolase super family, NADB_Rossmana super family and two UbiH genes in *B. brongniartii* region 2.1. Chiang et al. [42] found that an HR-PKS and an NR-PKS were adjacent to each other in the genome by studying 27 PKSs of *Aspergillus nidulans* (Eidam) G. Winter, which characterized the biosynthetic pathway of asperfuranone. The results showed that (Figure 2h) *L. fungicola* region 343.1 and its adjacent region 343.2 were highly like the gene cluster AACD01000015.1 that catalyzed the synthesis of asperfuranone. However, *L. fungicola* region 343.2 had not the SAT domain. KJ728786.1 catalyzed the synthesis of 4-epi-15-epi-brefeldin A. The core gene domain and modified genes of *B. brongniartii* region 25.2 (KS-AT-DH-ER-KR) and *L. fungicola* region 399.1 (KS-AT-DH-ER-KR-ACP) were similar to KJ728786.1 (Figure 2i), but there was not modification of the putative large tegument protein UL36 and putative WD-40 protein. *L. psalliotae* region 51.3 (Figure 2j) and *C. javanica* region (Figure 2k) were highly similar to BGCs synthesized from alternapyrone (AB120221.1) and solanapyrone D (AB514562.1), respectively. However, the core gene domain of *L. psalliotae* region 51.3 was missing DH and MT. von Bargen et al. [43]. found that the PKS19 gene cluster was a BGC for catalytic synthesis of fujikurins A−D, and through comparison, *S. hepiali* ICMM 82-2 region 4.4, *S. hepiali* FENG region 10.2, *S. yunnanensis* YFCC 1527 region 3.4, *L. fungicola* region 283.2 and *L. psalliotae* region 8.1 had a BGC for the synthesis of fujikurin A/fujikurin B/fujikurin C/fujikurin D (Figure 2l). The core gene domain of HF679030.1 was the KS-AT-DH-MT-KR-ACP, while *S. hepiali* ICMM 82-2 region 4.4, *S. hepiali* FENG region 10.2, *S. yunnanensis* YFCC 1527 region 3.4 and *L. fungicola* region 283.2 had added the Te domain. *L. psalliotae* region 8.1 was the KS-AT-KR-ACP. The GAL4 gene was absent in *S. hepiali* ICMM 82-2 region 4.4, *S. hepiali* FENG region 10.2, *L. fungicola* region 283.2 and *L. psalliotae* region 8.1, as well as *L. psalliotae* region 8.1 also had not the DLH super family. Meanwhile, the CYP gene was not found in *S. hepiali* ICMM 82-2 region 4.4, *S. hepiali* FENG region 10.2 and *L. fungicola* region 283.2. It was further indicated that different species in the same family had differences in some domains of PKS and the deletion or addition of modified genes.

We also found five compounds catalyzed by hybrid PKS-NRPS. *S. hepiali* ICMM 82-2 region 13.2, *S. hepiali* FENG region 11.2, *S. yunnanensis* YFCC 1527 region 32.1, *B. bassiana* region 4.4, *B. brongniartii* region 19.1 and *C. fumosorosea* region 34.2 had highly fumosorinone BGC-like gene clusters (Figure 2m). LC208781.1 was the BGC that catalyzed the synthesis of curvupallide-B. *S. hepiali* ICMM 82-2 region 16.2, *S. hepiali* FENG region 14.1, *S. yunnanensis* YFCC 1527 region 3.1 and *G. kalimantanensis* region 37.1 had gene clusters that were highly similar to synthetic curvupallide-B, and all had the core structural domain of KS-AT-DH-MT-KR-P-C-A-(P)-Te and modifier genes (Figure 2n). *C. javanica* region 7.2, *L. fungicola* region 21.1 and region 21.2 had BGC similar to synthetic phomasetin (Figure 2o). The core genes PKS and NRPS in *L. fungicola* region 21.1 and region 21.2 were separated by a gene whose function was unknown. *S. hepiali* FENG region 4.4, *C. cicadae* region 27.1, and *C. fumosorosea* region 5.3 had gene regions like those responsible for synthesizing wortmanamide A/wortmanamide B (Figure 2p). *A. lecanii* region 25.1 was similar to the gene cluster synthesizing aspyridone A (Figure 2q), both with the core structural domain of KS-AT-DH-MT-KR-P-C-A-P-Te and the modifier genes of CYP, Enoyl_reductase_like, PRK06126 super family, the Dehydrogenase, Fungal_TF_MHR, MFS_MCT_SLC016 and GAL4. The variability in the expression of the same compound in terms of the PKS or NRPS structural domains, modifier gene deletions and additions between genera of Cordycipitaceae, between species and between different strains of the same species was further demonstrated.

### 3.4. Synteny Analysis of 13 Species of Cordycipitaceae

The scaffolds containing the SM BGCs in the 14 genomes of Cordycipitaceae were subjected to synteny analysis. The scaffolds of the SM BGC were divided into 15 collinear blocks, and there might be rearrangements (Figure 3). From top to bottom, they were *A. lecanii*, *B. bassiana*, *B. brongniartii*, *B. pseudobassiana*, *C. cicadae*, *C. fumosorosea*, *C. javanica*, *G. kalimantanensis*, *L. fungicola*, *L. psalliotae*, *S. hepiali* FENG, *S. hepiali* ICMM 82-2 and *S. yunnanensis* YFCC 1527, respectively.

### 3.5. Cluster Analysis

#### 3.5.1. SM BGCs Cluster Analysis

The clustering results of the *Cordyceps* NRPS proteins with other fungal NRPS proteins showed that *L. fungicola* region 241.1 and *A. lecanii* region 4.5 converged with *A. terreus* (EAU36744.1), which catalyzed acetylaranotin biosynthesis (Figure 4a). We speculate that *L. fungicola* region 241.1 and *A. lecanii* region 4.5 may catalyze the biosynthesis of acetylaranotin or its analogists. *B. bassiana* region 3.4 and *B. brongniartii* region 15.1 converged with the NRPS protein in *B. bassiana* (ACR78148.1), which might catalyze the synthesis of basinolide. We suppose that the products catalyzed by *B. bassiana* region 3.4 and *B. brongniartii* region 15.1 may be basinolide or its analogues. *S. hepiali* ICMM 82-2 region 25.1, *S. hepiali* FENG region 6.2, *S. yunnanensis* YFCC 1527 region 11.2, *B. bassiana* region 8.4, *B. pseudobassiana* region 8.1, *L. fungicola* region 126.1, *L. psalliotae* region 95.2, *G. kalimantanensis* region 77.1, *C. javanica* region 1.1, *B. brongniartii* region 6.4, *C. fumosorosea* Region 75.1, *C. cicadae* region 56.2, as well as *A. lecanii* region 2.4, and *Alternaric alternata* (AFN69082.1) clustered on a separate branch, the product of which was dimethylcoprogen. Presumably, *S. hepiali* ICMM 82-2 region 25.1, *S. hepiali* FENG region 6.2, *S. yunnanensis* YFCC 1527 region 11.2, *B. bassiana* region 8.4, *B. pseudobassiana* region 8.1, *L. fungicola* region 126.1, *L. psalliotae* region 95.2, *G. kalimantanensis* region 77.1, *C. javanica* region 1.1, *B. brongniartii* region 6.4, *C. fumosorosea* region 75.1, *C. cicadae* region 56.2, as well as *A. lecanii* region 2.4 might catalyze the synthesis of dimethylcoprogen or its analogs. *A. lecanii* region 3.1 and *C. javanica* region 18.2 converged with *Epichloe festucae* (AET13875.1), which catalyzed the synthesis of epichloenin A. We speculated that epichloenin A or its analogized compound might be synthesized from *A. lecanii* region 3.1 and *C. javanica* region 18.2.

The cluster analysis results of the PKS and hybrid PKS-NRPS proteins of *Cordyceps* and other fungi showed that *A. lecanii* region 25.1 was clustered with the catalytic synthesis of aspyridone A (CBF80487.1) (Figure 4b), and we infer that the product catalyzed by *A. lecanii* region 25.1 may be aspyridone A or its analogue. *S. hepiali* ICMM 82-2 region 16.2, *S. hepiali* FENG region 14.1, *S. yunnanensis* YFCC 1527 region 3.1 and *G. kalimantanensis* region 37.1 were gathered in a branch with *Curvularia pallescens*, which catalyzed curvupallide-B biosynthesis. *S. hepiali* ICMM 82-2 region 16.2, *S. hepiali* FENG region 14.1, *S. yunnanensis* YFCC 1527 region 3.1 and *G. kalimantanensis* region 37.1 were speculated to catalyze curvupallide-B and its analogues. *S. hepiali* ICMM 82-2 region 13.2, *S. hepiali* FENG region 11.2, *S. yunnanensis* YFCC 1527 region 32.1, *B. bassiana* region 4.4, *B. brongniarti* region 19.1, and *C. fumosorosea* region 34.2 were combined with *C. fumosorosea* (AKC54422.1) to catalyze the synthesis of fumosorinone. We speculate that *S. hepiali* ICMM 82-2 region 13.2, *S. hepiali* FENG region 11.2, *S. yunnanensis* YFCC 1527 region 32.1, *B. bassiana* region 4.4, *B. brongniarti* region 19.1, and *C. fumosorosea* region 34.2 may catalyze the synthesis of fumosorinone or its analogues. *C. javanica* region 7.2 and *L. fungicola* region 21.1–21.2 converged with *Pyrenochaetosis* sp. (BBC43184.1) on an independent branch, and the protein sequence of *Pyrenochaetosis* sp. catalyzed the biosynthesis of phosphotin. We guess that *C. javanica* region 7.2 and *L. fungicola* region 21.1–21.2 may catalyze the synthesis of phomasetin or its analogues. *S. hepiali* FENG region 4.4, *C. fumosorosea* region 5.3, and *C. cicadae* region 27.1 converged on a separate branch with *Talaromyces wortmannii* (QBC19710.1), which catalyzed the biosynthesis of wortmanamide A/wortmanamide B. We conjecture that *S. hepiali* FENG region 4.4, *C. cicadae* region 27.1 and *C. fumosorosea* region 5.3 may catalyze the synthesis of wortmanamide A/ wortmanamide B or the biosynthesis of its analogues. *B. brongniartii* region 25.2 and *L. psalliotae* region 399.1 with *Penicillium brefeldianum* (AIA58899.1) might catalyze the synthesis of 4-epi-15-epi-brefeldin A PKS protein on a separate branch. We infer that the product catalyzed by *B. brongniartii* region 25.2 and *L. fungicola* region 399.1 may be 4-epi-15-epi-brefeldin A or its analogue. *L. psalliotae* region 51.3 got together with *Alternaria solani* (BAD83684.1), which catalyzed the synthesis of alternapyrone. We speculate that *L. psalliotae* region 51.3 may catalyze the synthesis of alternapyrone or its analogues. *L. fungicola* region 343.1 and *L. fungicola* region 343.2 were clustered in a branch with EAA65604.1 and EAA65602.1 of *A. nidulans*, respectively. EAA65604.1 and EAA65602.1 catalyze the synthesis of asperfuranone, and we guess that regions 343.1 and 343.2 of *L. fungicola* jointly catalyze the biosynthesis of asperfuranone or its analogues. *B. brongniartii* region 2.1 was clustered on a branch with *Fusarium fujikuroi* (CCT6799.1), a PKS protein sequence catalyzed for the synthesis of bikaverin. We suppose that bikaverin or its analogue is catalyzed by *B. brongniartii* region 2.1. *A. lecanii* region 5.5 is associated with *Monascus ruber* (ALI92655.1), which catalyzes the synthesis of citrinin, suggesting that *A. lecanii* region 5.5 may catalyze the synthesis of citrinin or its analogs. *S. hepiali* ICMM 82-2 region 4.4, *S. hepiali* FENG region 10.2, *S. yunnanensis* YFCC 1527 region 3.4, *L. fungicola* region 283.2 and *L. psalliotae* region 8.1 were clustered with *F. fujikuroi* (CCT72377.1) on a separate branch, the product of which was fujikurin A/fujikurin B/fujikurin C/fujikurin D. We hypothesize that *S. hepiali* ICMM 82-2 region 4.4, *S. hepiali* FENG region 10.2, *S. yunnanensis* YFCC 1527 region 3.4, *L. fungicola* region 283.2 and *L. psalliotae* region 8.1 may catalyze the synthesis of fujikurin A-D. *C. javanica* region 28.1 was clustered on an independent branch with *A. solani* (BAJ09789.1), which catalyzed the synthesis of solanapyrone D. We speculate that solanapyrone D is catalyzed by *C. javanica* region 28.1. *C. cicadae* region 117.1 and *C. fumosorosea* region 58.1 cluster on a branch with *Paecilomyces variotii* (XP_028481820.1). XP_028481820.1 is a PKS protein catalyzed to produce viriditoxin. Viriditoxin or its analogue could be synthesized from *C. cicadae* region 117.1 and *C. fumosorosea* region 58.1.

#### 3.5.2. Comparative Analysis of HR-PKS Homologous Region Cluster Tree and Multi-Gene Genetic Distance Tree

Latkowska et al. [44] suggested that the newly identified thalli compound from *Hypogymnia physodes* (L.) Nyl. might belong to its invariant or auxiliary SMs and could be used in the chemotaxonomic classification of this species. The classification of *Cordyceps* mainly relied on five gene data and mitochondrial genome comparison to assist with species identification. By comparing the genetic distance tree (Figure 5a) established based on polygenic data from 12 species of the Cordycipitaceae family with the cluster analysis tree established based on nucleic acid sequences containing the KS-AT-DH-ER-KR-ACP domain (Figure 5b), the results showed that the homologous sequence clustering trees with the same HR-PKS had the same shape as phylogenetic trees based on multi-gene data. We conjecture that PKS homologous sequences obtained through gene mining could be used as molecular markers to distinguish species.

## 4. Discussion

It had been shown that beauvericin (BEA) was a known cyclic hexapeptide compound in *B. bassiana* [45,46]. *B. bassiana* region 2.1 and *C. cicadae* region 87.1 were highly similar to the BEA gene clusters characterized by Xu et al. [47]. Zhang et al. [12] also indicated that the BEA was isolated from *C. cicadae*, and the of *C. cicadae* region 87.1 might be a BGC for catalytic synthesis of the BEA. We guess that the *B. bassiana* region 2.1 and the *C. cicadae* region 87.1 and their respectively modified genes are gene clusters that catalyze the synthesis of the BEA or its analogs. AntiSMASH predicted that *Simplicillium aogashimaense*, *C. cicadae*, and *C. fumosorosea* had the potential to catalyze the synthesis of viriditoxin, although further local BLSAT and cluster analysis showed that *Simplicillium aogashimaense* region 14.4 did not have the ability to catalyze the synthesis of viriditoxin. It was speculated that three ACP domains and SDR_c and Abhydrolase super family modified genes were required for the synthesis of viriditoxin. The BGCs predicted from *C. javanica* predicted biosynthesis genes that catalyze the synthesis of fumonisin.

In this study, gene mining was used to analyze whole-genome data from 14 genomes of 13 Cordycipitaceae species. The modified genes between *B. bassiana* regions 5.3 and 5.4, *B. brongniartii* regions 1.3 and 1.4, and *C. javanica* regions 1.5 and 1.6, as well as two regions of the three species, were PKS, acyltransferase, ATP-dependent long-chain fatty acyl-CoA synthetase and NRPS. This was consistent with the discovery of the catalytic synthesis of beauveriolide gene clusters in the *C. militaris* genome [11]. We previously hypothesized that *B. bassiana*, *B. brongniartii*, and *C. javanica* also have the potential to catalyze the synthesis of beauveriolide. AntiSMASH analysis predicted that *B. bassiana*, *B. brongniartii* and *Simplicillium aogashimaense* had the BGC for catalytic synthesis of bassianolide. The similarity to the known bassianolide gene clusters was 86%, 15.1% and 13%, respectively. Further BLAST analyses in the *B. bassiana* region 3.4, *B. brongniartii* region 15.1 and *Simplicillium aogashimaense* region 11.3 examined the potential for the catalytic synthesis of bassianolide. However, cluster analysis showed that the similarity between the structural domain and modified gene of *Simplicillium aogashimaense* region 11.3 and the known gene cluster of bassianolide synthesized by catalysis (ACR78148.1) was lower, and cluster analysis could not converge with the ACR78148.1. The bassianolide was synthesized catalytically from *B. bassiana* by cloning the encoding gene bbBsls [27]. We infer that *B. bassiana* region 3.4 and *B. brongniartii* region 15.1 could be the putative BGCs to produce bassianolide. The analysis revealed that *S. hepiali* ICMM 82-2 region 13.2, *S. hepiali* FENG region 11.2, *S. yunnanensis* YFCC 1527 region 32.1, *B. bassiana* region 4.4, *B. brongniartii* region 19.1 and *C. fumosorosea* region 34.2 had a BGC to catalyze the synthesis of fumosorinone. This greatly resembled the findings of Liu et al. [26], who characterized fumosorinone as a BGC catalyzed by the hybrid PKS-NRPS, two cytochrome P450, trans-enoyl reductase gene and two other transcriptions regulatory genes fumosorinone.

The results of the 14 genome-wide data analysis for 13 species of Cordycipitaceae indicated that *C. cicadae* region 87.1, region 56.2, region 27.1 and region 117.1 might exist to catalyze the synthesis of the BEA, dimethylcoprogen, wortmanamide A/B and viriditoxin, respectively, in agreement with our previous analysis of *C. cicadae* [28]. Among the 14 genomes, the potential gene clusters of acetylaranotin, bassianolide, dimethylcoprogen, epichloenin A, aspyridone A, curvupallide-B, fumosorinone, phomasetin, wortmanamide A/B, 4-epi-15-epi-brefeldin A, alternapyrone, asperfuranone, bikaverin, citrinin, fujikurin, solanapyrone D and viriditoxin were also excavated. Among them, the SM BGCs unique to *A. lecanii*, *L. psalliotae*, *L. fungicola*, *B. brongniartii* and *C. javanica* were aspyridone, alternapyrone, asperfuranone, bikaverin and solanapyrone D, respectively. We speculate there to be some species specificity in these five species. Therefore, we suppose that the number and type of the putative SM BGCs, the expression of the same compound in the PKS or NRPS domain, and the deletion and addition of modification genes differ greatly among different genera of Cordycipitaceae, different species of the same genus and different strains of the same species. These predictions of the putative SM BGC required further validation via heterologous expression or gene knockout. 

Scaffolds containing the SM BGCs in the genomes of 13 Cordycipitaceae species had been shown to be divided into 15 collinear blocks in 12 species, excluding *Simplicillium aogashimaense*. The scaffold with *Simplicillium aogashimaense* SM BGC does not have collinear blocks with 12 species. Presumably, the taxonomic position of *Simplicillium aogashimaense* was far removed from that of the other 12 species, and therefore, there was not a collinear block with the other 13 genomes. We compared the phylogenetic tree established using polygenic data with the cluster analysis tree established via NRPS (A-P-C), NR-PKS (SAT-KS-AT-PT-ACP-ACP-Te), HR-PKS (KS-AT-DH-ER-KR-ACP), and hybrid PKS-NRPS (KS-AT-DH-MT-KR-P-C-A-P-Te), respectively. We found that among the 13 species of Cordycipitaceae, only the ML tree and BI tree constructed by the HR-PKS homologous region are highly similar to the polygenic genetic distance tree. Due to the large number of heterogeneous homologous sequences of NRPS, NR-PKS, HR-PK and hybrid PKS-NRPS in the 13 species of Cordycipitaceae, the similarity between the cluster analysis trees established and the polygenic genetic distance was lower. *G. kalimantanensis* was not used in the polygenic phylogenetic tree due to the lack of five gene data. Further expansion of the Cordycipitaceae species is needed to verify whether the PKS homologous sequences obtained via gene mining can be used as molecular markers to distinguish species.

## 5. Conclusions

Analysis of 14 genomes of 13 species in 7 genera of Cordycipitaceae showed that there were significant differences in the genome size, GC content, number and type of the putative SM BGCs, NPPS domain, PKS domain and hybrid PKS-NRPS domain among species of different genera, different species of the same genus, and different strains of the same species.

The specific domain of *S. hepiali* ICMM 82-2, *S. hepiali* FENG and *S. yunnanensis* YFCC 1527 was shown to be the SAT-KS-AT-PT-ACP via the antiSMASH and local BLAST analysis. The unique structural domains of *S. hepiali* ICMM 82-2 were the A-C-P-C-P-C, AT-DH-MT-KR, KS-AT-DH and KR-ACP-C-A-Te. The specific domains, such as A-P-C-P-P-C, A-P-A-C-A-P, KS-AT-DH-ER-KR-ACP-ACP, KS-AT-DH-MT-ER-ACP-ACP, KS-AT-DH-MT-KR-ACP-C-A-Te, KS-AT-DH-ER-KR-P-C-A-Te, A-P-KS-AT-KR-ACP, KR-P-C-A-Te and A-P-Te-KR were located in the *S. hepiali* FENG. The unique domain of *S. hepiali* was SAT-KS-AT-PT-ACP-ACP. The special domains, such as the C-A-A-A-P-C-A-P-C, C-A-A-P-C-A-Te, KS-AT-DH-MT-KR-ACP, AT-ACP-Aes, SAT-KS-PT-ACP-ACP, and A-P-KS-AT, were located in *S. yunnanensis* YFCC 1527.

All 14 genomes had the special domains A-P-C-P-C and KS-AT-DH-ER-KR-ACP. In particular, the A-P-C-P-C domain might be the key enzyme domain of the putative BGC for the catalytic synthesis of dimethylcoprogen, while the ML tree and the BI tree with the KS-AT-DH-ER-KR-ACP homologous regions were extremely similar to the trees of polygenic phylogeny. The curvupallide-B, fumosorinone and fujikurin putative BGCs were all present in *S. hepiali* ICMM 82-2, *S. hepiali* FENG and *S. yunnanensis* YFCC 1527. In addition, 13 putative SM BGCs were found to be highly similar to known gene clusters, suggesting that the Cordycipitaceae species might have great potential for SM production. At the same time, the scaffold of potential gene clusters was divided into 15 collinear blocks in the 13 genomes of the 12 species in the Cordycipitaceae family.

## Figures and Tables

**Figure 1 jof-09-01146-f001:**
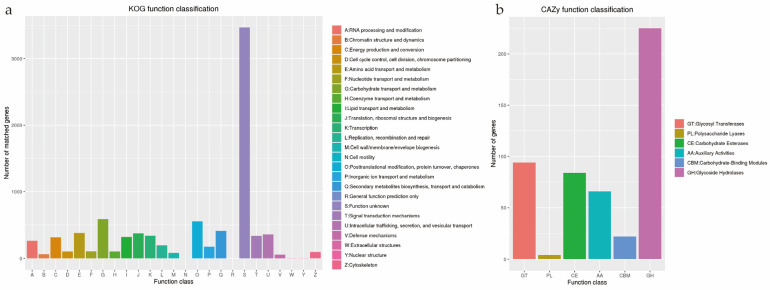
Functional annotation of *S. hepiali* ICMM 82-2 genes encoding the proteins: (**a**) eggNOG analysis; and (**b**) CAZy analysis. The vertical coordinates indicate the number of genes (left), and the horizontal coordinates indicate the annotated functional genes involved in the biological function categories.

**Figure 2 jof-09-01146-f002:**
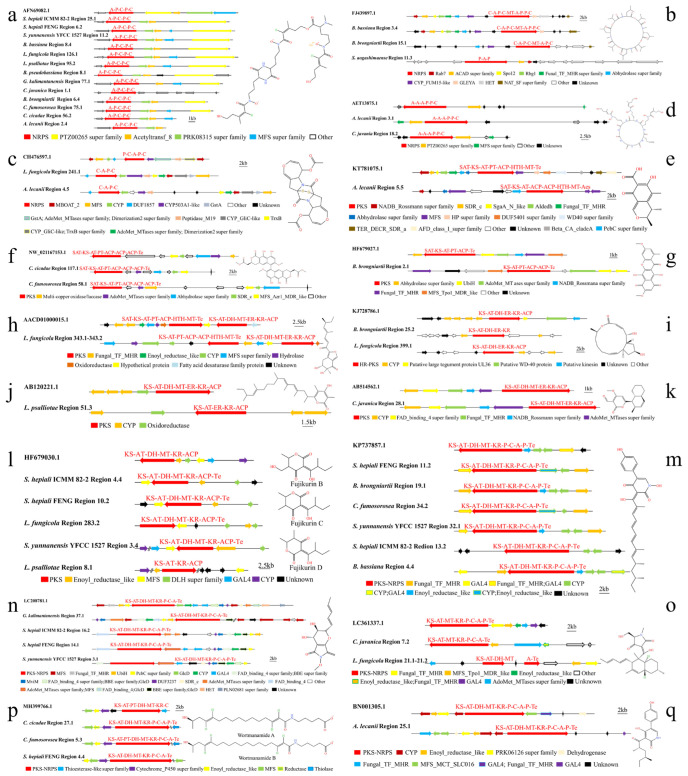
Comparison of biosynthesis of putative dimethylcoprogen (**a**), bassianolide (**b**), acetylaranotin (**c**), epichloenin A (**d**), citrinin (**e**), viriditoxin (**f**), bikaverin (**g**), asperfuranone (**h**), 4-epi-15-epi-brefeldin A (**i**), alternapyrone (**j**), solanapyrone D (**k**), fujikurin A/B/C/D (**l**), fumosorinone (**m**), curvupallide-B (**n**), phomasetin (**o**), wortmanamide A/B (**p**), and aspyridone A (**q**). The number after the region and the number before the decimal point represent the scaffold, and the number after the decimal point represents the gene cluster.

**Figure 3 jof-09-01146-f003:**
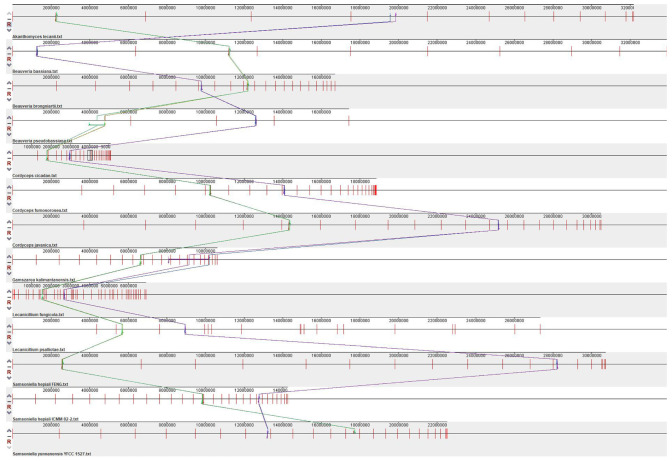
Scaffold synteny analysis of biosynthetic gene clusters containing secondary metabolites in the genomes of 13 genomes of Cordycipitaceae. Numerical representation of gene length (bp). Different color lines represent collinear regions between different genomes.

**Figure 4 jof-09-01146-f004:**
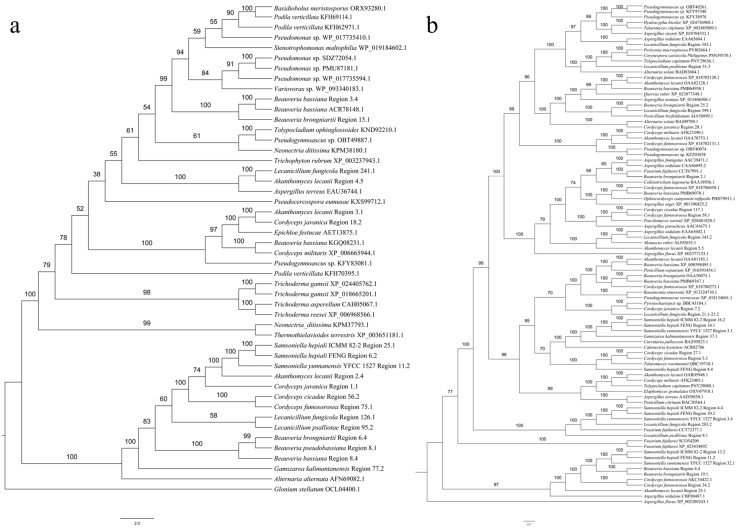
Clustering ML tree of NRPS/PKS/hybrid PKS-NRPS and other fungal NRPS/PKS/hybrid PKS-NRPS proteins in the fourteen genomes. Values at the nodes represent bootstrap values. The scale bars 2.0 (**a**) and 0.7 (**b**) indicate the number of expected mutations per site. *Glonium stellatum* (**a**) and *Aspergillus flavus* (**b**) are used as the outgroup. (**a**) NRPS cluster analysis tree; (**b**) PKS/hybrid PKS-NRPS cluster analysis tree.

**Figure 5 jof-09-01146-f005:**
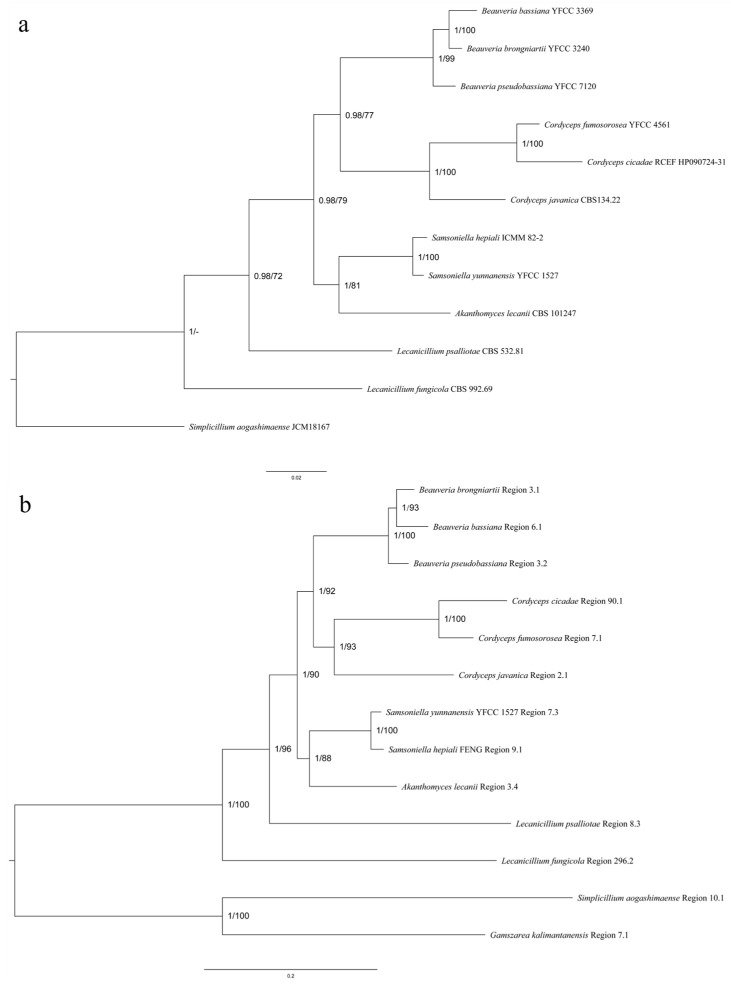
Comparison between multigene phylogenetic tree and cluster analysis tree containing KS-AT-DH-ER-KR-ACP homologous genes. Values at the nodes represent the ML bootstrap proportions (**a**) and the BI posterior probabilities (**b**). All values are shown at the nodes. The scale bars 0.02 (**a**) and 0.2 (**b**) indicate the number of expected mutations per site. *Simplicillium aogashimaense* (**a**) and *Gamszarea kalimantanensi* (**b**) are used as the outgroup. (**a**) The phylogenetic tree of 12 fungi multi-gene data based on the ML and the BI analyses. (**b**) The 13 fungi studied contained KS-AT-DH-ER-KR-ACP structure clustering analysis tree (the ML and the BI trees).

## Data Availability

The data presented in this study are available in the Supplementary Material. The genome of *S. hepiali* ICMM 82-2 and *S. yunnanensis* YFCC 1527 sequence was uploaded to the SRA database (https://submit.ncbi.nlm.nih.gov/subs/sra/) (accessed on 12 November 2023). The *S. hepiali* ICMM 82-2 and *S. yunnanensis* YFCC 1527 genome data are available from SRA. The SRA ID, BioProject ID and BioSample ID of *S. hepiali* ICMM 82-2 was SRR26591298, PRJNA1033426, SAMN38033584, respectively. The SRA ID, BioProject ID and BioSample ID of *S. yunnanensis* YFCC 1527 was SRR26593767, PRJNA1033999, SAMN38048170, respectively.

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
