# Peer review of "Genomic Comparison of Two Species of Samsoniella with Other Genera in the Family Cordycipitaceae"

_jof, 2023, doi:10.3390/jof9121146_

Round 1
Reviewer 1 Report
Comments and Suggestions for Authors
Dear authors,
The manuscript is relevant and presents new metabolites produced by fungi from the Cordycipitaceae family. As a way of contributing to the improvement of the text, here are some observations for evaluation:
The introduction is well-written and adequately introduces the motivation for the work. The literature used is appropriate and relevant to the topic. However, the introduction presents an unnecessary excess of information, which gives it the appearance of a bibliographic review. I understand the authors' care, but I think a review is important to eliminate information that does not add to the text. An example that stands out is the presentation of "Figure 1", as it does not add any relevant information to the text.
The topic material and methods are robust, using methodologies widely reported in the literature. Likewise, a review of the text is pertinent, as there is an excess of unnecessary information. An example is linked to the topic “2.1. Test materials”. Just enter the geographic coordinates of the collection location. Excessive unnecessary information can overshadow the quality of the work.
The results are suitable for the proposed methodology, however, I recommend a revision of the text, to make the information more streamlined.
The discussion is appropriate and presents literature relevant to the topic. In addition to the observations made above, I recommend a review of the language, as some words were written incorrectly. Congratulations on the award.
Author Response
Dear reviewer
Sincerely thanks for your valuable suggestions. We realized that we should have a more rigorous attitude towards paper. The manuscript has been revised according to your comments, and the answers are as follows: (marked with color yellow in the manuscript)
Comment 1: The introduction is well-written and adequately introduces the motivation for the work. The literature used is appropriate and relevant to the topic. However, the introduction presents an unnecessary excess of information, which gives it the appearance of a bibliographic review. I understand the authors' care, but I think a review is important to eliminate information that does not add to the text. An example that stands out is the presentation of "Figure 1", as it does not add any relevant information to the text.
Response 1: Thank you very much for your suggestion! The introduction did state a lot of superfluous information and has now been modified. Figure 1 has been removed, see line 49.
Expression has been modified to “Cordycipitaceae species produced a variety of metabolites, such as bassianolide [4], fumosorinone [5], ergosterol [6], 8-O-methylindigotide B [7], cordycerebroside B [8], cepharosporolides C [9], beauvericin [10], beauveriolide [11], oosporein [12], bassiatin [13], ophiocordin [14], cordycepin [15], oosporein [16] and cordycepic acid [17]. It could be seen that there is still a lot of room for exploration of the potential of Cordycipitaceae species to produce compounds.”, “The Secondary Metabolites Analysis Shell (antiSMASH) allows for rapid and direct detection of biosynthetic gene clusters (BGCs), as well as the diversity of gene cluster families.”, “Some also have domains such as keepeduktase (KR), dehydratase (DH), enoylreduktase (ER), and methyltransferase (MT) to explain the large number of structurally complex and diverse metabolites available through this common pathway [20, 21]. The domain of NRPS is usually composed of adenylation (A) and condensation (C)”, and “More SM BGCs can be discovered through gene mining. The amino acid sequence of NpPKS3 in the Lichenized-fungi Nephrmopsis pallescensi was 53% consistent with the Type III PKS (CYSBs) of B. bassiana”. (please see the Revised Manuscript, lines 44-49, 52-54, 58-62, and 64-66)
Comment 2: The topic material and methods are robust, using methodologies widely reported in the literature. Likewise, a review of the text is pertinent, as there is an excess of unnecessary information. An example is linked to the topic “2.1. Test materials”. Just enter the geographic coordinates of the collection location. Excessive unnecessary information can overshadow the quality of the work.
Response 2: Thanks for this suggestion! This was due to a misunderstanding on our part. It has been modified to “For this study, the strain ICMM 82-2 of S. hepiali and YFCC 1527 of S. yunnanensis were collected from Yunnan Province in China” in 2.1. Test materials (please see the Revised Manuscript, lines 89-90).
Comment 3: The results are suitable for the proposed methodology, however, I recommend a revision of the text, to make the information more streamlined.
Response 3: Many thanks for your suggestion! The results section has been revised to make it more streamlined. Expression has been modified to “The result of the annotation of PHI showed that the major annotated genes of S. hepiali ICMM 82-2 and S. yunnanensis YFCC 1527 PHI-base decreased virulence and did not affect pathogenicity, indicating that S. hepiali ICMM 82-2 (Fig. S6) and S. yunnanensis YFCC 1527 (Fig. S7) were not highly pathogenic strains.”, “The results showed that S. hepiali ICMM 82-2 (Fig. 1b) and S. yunnanensis YFCC 1527 (Fig. S8) had a lot of glycoside hydrolases (GHs), glycosyl transferases (GTs), and sauxiliary activities (AAs), hypothesizing that S. hepiali ICMM 82-2 and S. yunnanensis YFCC 1527 might have the ability to capture more energy and decompose complex carbohydrates.”, “Gamszarea and Simplicillium, which were relatively distant relatives of Samsoniella, also showed differences between the results of genomic analyses of these two species and those of the three species of Samsoniella. It could be seen that the number and type of putative SM BGC varied considerably between species of different genera and be-tween species of the same genus.”, and “The analysis results of 14 genomes of Cordycipitaceae (Table S4) showed that there were also significant differences in the structural domains of PKS, NRPS and hybrid PKS-NRPS. The domains A-P-C-P-C and KS-AT-DH-ER-KR-ACP were shared by all 14 genomes, and the A-P-T domain was shared by all 13 genomes except Simplicillium aogashimaense. Only C. javanica had hybrid PKS-NRPS-Other of the KS-AT-MT-P-P-C-A-P-Te-PX-Snx8 domain. Except for S. hepiali ICMM 82-2 and C. fumosorosea, all the other genomes had the A-C-P-C-P-C domain. Only A. lecanii, C. fumosorosea, and L. psalliotae had not the KS-AT-DH-MT-ER-KR-ACP domain. The KS-AT-DH-ER-KR-P-C-A-P-Te domain existed only in the genomes of S. hepiali ICMM 82-2, S. yunnanensis YFCC 1527, A. lecanii, C. fumosorosea, L. psalliotae, and G. kalimantanensis”. (please see the Revised Manuscript, line 184-188, 193-197, 230-234, and 244-253).
Comment 4: The discussion is appropriate and presents literature relevant to the topic. In addition to the observations made above, I recommend a review of the language, as some words were written incorrectly. Congratulations on the award.
Response 4: Thanks for this suggestion! This was due to our mistake. Expression has been modified to “Simplicillium” (please see the Revised Manuscript, lines 41, 95, 205, 279, 283, 309, 312, 482, 484, 497, 499, 502, and 530-532), “B. brongniarti” (please see the Revised Manuscript, lines 379 and 382), and “B. bassiana” (please see the Revised Manuscript, lines 379-381, 412, and 416).
Best regard!
Hong Yu
Reviewer 2 Report
Comments and Suggestions for Authors
Lu et al. have compared the genomes of two new Samsoniella strains with 12 publicly available genome sequences from the fungal family Cordycipitaceae, focusing on similarities and differences in the genomic basis of secondary metabolite production potentials of the analyzed strains. They present an enormous bulk of comparative data. The manuscript is highly relevant and there is no doubt that it deserves publication in JoF.
However, presentation of the data has to be improved. Throughout the manuscript there is an issue with English language, professional editing might be required.
Some points to be improved:
Suppl Materials: please provide a separate PDF file for each Table.
Results section: it is not useful to enumerate the data obtained in the text, parts of the manuscript as, e.g., genome annotation section (3.1.2) are hardly legible. Please try to organize the data in (Suppl) Tables and only cite particularly conspicuous pieces of data in the text. A similar issue is with Figure 2: authors should decide which graphs are the most relevant to be shown in the main text and provide others as Suppl Mat.
Figure legends: in general, Figure legends could be more informative. For instance Figures 5 and 6: What does numbers on branches mean? Has there been a technical outgroup used in tree construction? What does the size bar – if present – indicate?
Citations in the text should be given in brackets, not as superscript.
First paragraph of introduction might start in present tense as what is stated is current knowledge and still true today, such as: “Cordycipitaceae (Ascomycota: Hypocreales) is a family comprising parasitic fungi including 25 genera such as Cordyceps, Beauveria, Akanthomyces, and Simplicillium.” or similar. Same for following sentences.
There is an issue with the oprthography of taxon designations across the manuscript, please check carefully:
Simplicillium, not Simplicidium (line 41, 120 and further)
B. brongniarti, not B. bronniarti
B. bassiana, not B. basiana
Comments on the Quality of English Languagemoderate English language editing required, check use of capital letters, syntactic correctness of wording, standards for enumeration.
Author Response
Responses to the reviewer comments
Dear reviewer
Thank you very much for your insightful and helpful comments on our manuscript. We have revised the paper according to your comments. Our response to the reviewers’ suggestion is as follows: (marked with color green in the manuscript)
Comment 1: However, presentation of the data has to be improved. Throughout the manuscript there is an issue with English language, professional editing might be required.
Response 1: Thank you very much for your advice! Changes have been made to the way the data is presented. I invited a foreign friend to edit and modify the English language
Comment 2: Suppl Materials: please provide a separate PDF file for each Table.
Response 2: Thank you very much for your suggestion! A separate PDF file is provided for each table.
Comment 3: Results section: it is not useful to enumerate the data obtained in the text, parts of the manuscript as, e.g., genome annotation section (3.1.2) are hardly legible. Please try to organize the data in (Suppl) Tables and only cite particularly conspicuous pieces of data in the text. A similar issue is with Figure 2: authors should decide which graphs are the most relevant to be shown in the main text and provide others as Suppl Mat.
Response 3: Many thanks for your suggestion! The section on data and graphs have been revised. Only the EggNOg analysis and CAZy analysis of S. hepiali ICMM 82-2 are included in Figure 2. The rest of the diagrams have been included in the supplementary materials (Figure S1-S8) and each diagram is presented as a separate PDF file. (please see the Revised Manuscript, line 159, 166, 170, 187, and 194)
The following is the revised content related to data. The genome annotation section (3.1.2) expression have been modified to “The EggNOg database showed that most of the genes were predicted to have the general Function of S. hepiali ICMM 82-2 (Fig. 1a) and S. yunnanensis YFCC 1527 (Fig. S1), and the rest were in "Function unknown". Secondly, the “Carbohydrate transport and metabolism, Posttranslational modification, protein turnover, chaperones” were the most abundant gene class maps in the EggNOg group. It was indicated that S. hepiali ICMM 82-2 and S. yunnanensis YFCC 1527 had abundant and diverse carbohydrate metabolic functions, and posttranslational events. It may help to improve energy con-version efficiency and regulate protein bioactivity. The results of KEGG functional classification showed that S. hepiali ICMM 82-2 (Fig. S2) and S. yunnanensis YFCC 1527 (Fig. S3) had high activity of protein families, indicating that the two strains had a va-riety of genetic information, signals and cell proteins, and might have higher infor-mation exchange efficiency and secondary metabolic efficiency. According to GO an-notation results (S. hepiali ICMM 82-2: Fig. S4; S. yunnanensis YFCC 1527: Fig. S4), from the cell component category, intracellular and cellular component, from biological processes, cellular nitrogen compound metabolic process and the biosynthetic process, and molecular function from molecular function, ion binding and oxidoreductase ac-tivity. It was further indicated that S. hepiali ICMM 82-2 and S. yunnanensis YFCC 1527, as wild strains, might be related to signal transduction in many metabolic genes.” (please see the Revised Manuscript, line 158-175)
The overview of 14 genomic BGCs of Cordycipitaceae species section (3.3.1) expression have been modified to “A. lecanii had 46 SM BGCs and no Other gene.” and “C. cicadae, C. javanica and C. fumosorosea had 31, 51 and 54 putative SM BGCs, respectively, including the number of NRPS, PK, hybrid PKS-Other and Other gene were very different. The three species in Beauveria differ greatly only in the amount containing hybrid PKS-Othe.” (please see the Revised Manuscript, line 218-219, 227-230)
The synteny analysis of 13 species of Cordycipitaceae section (3.4) expression have been modified to “The scaffolds containing the SM BGCs in the 14 genomes of Cordycipitaceae were subjected to synteny analysis.” (please see the Revised Manuscript, line 364-365)
Comment 4: Figure legends: in general, Figure legends could be more informative. For instance Figures 5 and 6: What does numbers on branches mean? Has there been a technical outgroup used in tree construction? What does the size bar – if present – indicate?
Response 4: Many thanks for your valuable advice! Figures 5 and 6 have been changed to Figures 4 and 5 due to adjustments in content. The values on the branches in Figures 4 and 5 represented bootstrap values. The trees in Figures 4 and Figure 5 used the outer class group. The exomorphs in Figures 4a, 4b, 5a and 5b were Glonium stellatum, Aspergillus flavus, Simplicillium aogashimaense and Gamszarea kalimantanensis, respectively. Because Gamszarea kalimantanensis had no five-gene sequence, Simplicillium aogashimaense was used as an outgroup in Figure 5a. The scale bar 2.0 (Fig.4a), 0.7 (Fig.4b), 0.02 (Fig. 5a) and 0.2 (Fig. 5b) indicated the number of expected mutations per site, respectively.
The legend in Figure 1 have been modified to “Functional annotation of S. hepiali ICMM 82-2 genes encoding the proteins. a: EggNOg analysis; b: CAZy analysis. The vertical coordinates indicate the number of genes (left), and the horizontal coordinates indicate the annotated functional genes involved in biological function categories.” (please see the Revised Manuscript, line 177-180)
The legend in Figure 3 have been modified to “Scaffolds synteny analysis of biosynthetic gene clusters containing secondary metabolites in the genomes of 13 genomes of Cordycipitaceae. Numerical representation of gene length (bp).” (please see the Revised Manuscript, line 371-372)
The legend in Figure 4 have been modified to “Clustering ML tree of NRPS/PKS/hybrid PKS-NRPS and other fungal NRPS/PKS/hybrid PKS-NRPS proteins in the fourteen genomes. Values at the nodes represent bootstrap values. The scale bar 2.0 (a) and 0.7 (b) indicate the number of expected mutations per site. Glonium stellatum (a) and Aspergillus flavus (b) are used as the outgroup. a: NRPS cluster analysis tree; b: PKS/Hybrid PKS-NRPS cluster analysis tree.”. (please see the Revised Manuscript, line 398-402)
The legend in Figure 5 have been modified to “Comparison between multigene phylogenetic tree and cluster analysis tree containing KS-AT-DH-ER-KR-ACP homologous genes. Values at the nodes represent the ML bootstrap propor-tions (a) and the BI posterior probabilities (b). All values are shown at the nodes. The scale bar 0.02 (a) and 0.2 (b) indicate the number of expected mutations per site. Simplicillium aogashimaense (a) and Gamszarea kalimantanensi (b) are used as the outgroup. a: The phylogenetic tree of 12 fungi multigene data based on the ML and the BI analyses; b: The 13 fungi studied contained KS-AT-DH-ER-KR-ACP structure clustering analysis tree (the ML and the BI trees).” (please see the Revised Manuscript, line 468-474)
Comment 5: Citations in the text should be given in brackets, not as superscript.
Response 5: Thanks for this suggestion! This was owing to our misunderstanding. It has been modified to “Wang et al. [11] discovered beauveriolide …”, “…. had been discovered as putative BGCs [32]” and “… fungi and insects [33]”. (please see the Revised Manuscript, line 68, 81, 184)
Comment 6: First paragraph of introduction might start in present tense as what is stated is current knowledge and still true today, such as: “Cordycipitaceae (Ascomycota: Hypocreales) is a family comprising parasitic fungi including 25 genera such as Cordyceps, Beauveria, Akanthomyces, and Simplicillium.” or similar. Same for following sentences.
Response 6: Thank you very much for your valuable advice! This was due to a misunderstanding on our part. The first paragraph of introduction has been modified to “Cordycipitaceae (Ascomycota: Hypocreales) is a family comprising parasitic fungi including 25 genera such as Cordyceps, Beauveria, Akanthomyces, and Simplicillium [1-2]. Among them, Samsoniella hepiali (Q.T. Chen & R.Q. Dai ex R.Q. Dai et al.) H. Yu, R.Q. Dai, Y.B. Wang, et al. is an industrial strain for producing Jinshuibao, its related species is S. yunnanensis YFCC 1527 H. Yu, Y.B. Wang, Y. Wang et al [3]. Based on the current classification position, the genus and species of related S. hepiali are A. lecanii (Zimm.) Spatafora, Kepler & B. Shrestha, C. cicadae (Miq) Massee, C. javanica (Bally) Kepler, B. Shrestha & Spatafora, C. fumosorosea (Wize) Kepler, B. Shrestha & Spatafora, B. bassiana (Bals.-Criv.) Vuill., B. pseudobassiana S.A. Rehner & Humber, B. brongniartii (Sacc.) Petch, Lecanicillium fungicola (Preuss) Zare & W. Gams and L. psalliotae (Treschew) Zare & W. Gams, the farther genera species apart are Simplicidium aogashimaense Nonaka, Kaifuchi & Masuma and Gamszarea kalimantanensis (Kurihara & Sukarno) Z.F. Zhang & L. Ca [2].” (please see the Revised Manuscript, line 31-42). The following sentences adjust the tenses accordingly according to the knowledge. (please see the Revised Manuscript, line 53-54, 56, 58,61).
Comment 7: There is an issue with the oprthography of taxon designations across the manuscript, please check carefully:
Simplicillium, not Simplicidium (line 41, 120 and further)
- brongniarti, not B. bronniarti
- bassiana, not B. basiana
Response 7: Thanks for this suggestion! This was due to our mistake. Expression has been modified to “Simplicillium” (please see the Revised Manuscript, lines 41, 95, 205, 279, 283, 309, 312, 482, 484, 497, 499, 502, and 530-532), “B. brongniarti” (please see the Revised Manuscript, lines 379 and 382), and “B. bassiana” (please see the Revised Manuscript, lines 379-381, 412, and 416).
Best regard!
Hong Yu